# Association of Lower Nutritional Status and Education Level with the Severity of Depression Symptoms in Older Adults—A Cross Sectional Survey

**DOI:** 10.3390/nu13020515

**Published:** 2021-02-04

**Authors:** Zuzanna Chrzastek, Agnieszka Guligowska, Bartlomiej Soltysik, Malgorzata Piglowska, Ewa Borowiak, Joanna Kostka, Tomasz Kostka

**Affiliations:** 1Department of Geriatrics, Healthy Ageing Research Centre, Medical University of Lodz, 92-115 Lodz, Poland; zuzanna.chrzastek@umed.lodz.pl (Z.C.); agnieszka.guligowska@umed.lodz.pl (A.G.); bartlomiej.soltysik@umed.lodz.pl (B.S.); malgorzata.piglowska@umed.lodz.pl (M.P.); 2Department of Conservative Nursing, Medical University of Lodz, 90-251 Lodz, Poland; ewa.borowiak@umed.lodz.pl; 3Department of Gerontology, Medical University of Lodz, 93-113 Lodz, Poland; joanna.kostka@umed.lodz.pl

**Keywords:** depression, nutritional status, education level, older people

## Abstract

The study analyzes the relationship between nutritional status and depression symptoms severity in the older population. A total of 1975 older outpatients (1457 women and 518 men, median age 75) were included in the study. Depression symptoms severity was assessed using the Geriatric Depression Scale (GDS). Participants were divided into two subgroups according to GDS score. Group A: 0–5 points—without depression symptoms (1237, W:898, M:339), and group B: 6–15 points—with depression symptoms (738, W:559, M:179). The nutritional status of the patients was assessed with Mini Nutritional Assessment (MNA) and basic anthropometric variables (waist, hips, calf circumferences, body mass index (BMI), waist to hip ratio (WHR), and waist to height ratio (WHtR)). Education years and chronic diseases were also noted. Women with higher depression symptoms severity had significantly lower MNA scores [A: 26.5 (24–28) (median (25%−75% quartiles)) vs. B:23 (20.5–26)], shorter education time [A:12 (8–16) vs. B:7 (7–12)], smaller calf circumference [A:36 (33–38) vs. B: 34 (32–37)], and higher WHtR score [A:57.4 (52.3–62.9) vs. B:58.8 (52.1–65.6)]. Men with depression symptoms had lower MNA scores [A:26.5 (24.5–28) vs. B:24 (20.5–26.5)], shorter education [A:12 (9.5–16), B:10 (7–12)], and smaller calf circumference [A:37 (34–39), B:36 (33–38)]. In the model of stepwise multiple regression including age, years of education, anthropometric variables, MNA and concomitant diseases nutritional assessment, and education years were the only independent variables predicting severity of depression symptoms both in women and men. Additionally, in the female group, odds were higher with higher WHtR. Results obtained in the study indicate a strong relationship between proper nutritional status and education level with depression symptoms severity in older women and men.

## 1. Introduction

Recent decades have shown an increase in the proportion of older adults in general population due to extended lifespans. The World Health Organization (WHO) estimates that the population of people aged 60 + will double to 22% from 2015 to 2050 [1]. Ageing is characterized by multiple diseases and health issues and a reduction in the functionality of the organism as a whole, alongside a visible and progressive decline in cognitive functions. Mental or neurological disorders affect up to 20% of people over 60 worldwide. According to the WHO, one of the most common chronic psychological conditions is depression affecting as many as 264 million people worldwide and as much as 7% of the population over 60 [1,2].

Deterioration of physical and mental health in older patients is often associated with inadequate diet and lifestyle [3]. The impact of diet and its individual nutrients on the risk of depression and the severity of its symptoms has been noted in previous studies [4,5]. Older people are particularly vulnerable to improper nutritional status, which may result in deterioration of bodily functions caused by deficiency of nutritional agents [6]. Malnutrition can aggravate existing problems and contribute to development of new ones [7,8]. Nutritional status is considered to be one of the predictors of survival in the older population [9]. Malnutrition is a very common problem in older patients: it is estimated that over 50% of older adults are malnourished or at risk of malnutrition; however, these calculations depend on the tool used for assessment, studied population, and healthcare quality [10,11]. In addition, seniors from all backgrounds, i.e., home, hospital, and nursing home, are at risk of malnutrition [12].

It seems that malnutrition can lead to development of depression symptoms and, conversely, existing depression may affect nutritional problems, reluctance to eat, and thus, lead to development of malnutrition [13,14]. Older, malnourished persons were 31% more likely to present symptoms of depression than people with normal nutritional status [15]. Introduction of an appropriate intervention addressing each of those disorders appears to improve the effectiveness of treatment [16].

Previous studies have examined small groups or have failed to assess a connection between basic anthropometric variables and indicators of nutritional status in the context of depressive disorders. Few studies concern older people population. Likewise, education level and commonly found in older patients’ concomitant diseases may have an impact on the nutritional status-depression relationship. Therefore, the aim of this study was to indicate which variables: education years, anthropometric indicators, nutritional status, and concomitant diseases may have the strongest relation with depression symptoms. So far, no large research has been carried out to clarify the relationship between these variables and depression symptoms severity.

## 2. Materials and Methods

### 2.1. Design of the Study and Participants

Study participants were community-dwelling older adults. Their visits to the Geriatric Clinic were conditioned not only by health check-up visits but also for sole participation in the research projects conducted in the clinic. Predominance of women in the studied group results from the demographic profile of the Polish population. The ratio of males to females is 0.66 in age group ≥65, reflecting a higher mortality rate among older men [17,18]. Another factor influencing superior number of women over men included in the study may be the fact that women view their quality of life and health as worse as shown in the PolSenior - Study on Ageing and Longevity, therefore, are keener to participate in general health checkup studies [19]. The inclusion criteria were as follows: 60 years of age or older, ability to intake food orally, lack of communication, and comprehension problems as well as informed consent for participation in the study. Patients with severe dementia or those receiving enteral nutrition were excluded from the study. Depression symptoms severity was assessed with the 15-item Geriatric Depression Scale (GDS) questionnaire. The GDS is one of the most popular self-reporting tests used to assess the presence of depression symptoms in the older population [20]. It includes 15 questions with simple yes/no alternative answers. A higher GDS score indicates greater severity of depression symptoms: the highest possible score is 15 and obtaining five points or less indicates absence of depression [21]. In our study, patients were divided into two subgroups according to GDS score. Group A included those with a lack of depression symptoms (0–5 points): 1237 patients (898 women and 339 men). Group B included patients with depression symptoms (> 5 points): 738 patients (559 women and 179 men).

### 2.2. Procedure

A comprehensive geriatric assessment was conducted during face-to-face interviews by qualified researchers including medical doctors, nutritionists, and PhD students from the Geriatric Clinic of the Medical University of Lodz. All participants underwent physical and mental examinations. Nutritional status was assessed using the Mini Nutritional Assessment (MNA) questionnaire, which contains 18 questions relating to important elements of nutritional status including food intake, weight loss, mobility, presence of acute stress or disease, neurological problems, intake of medications, body mass index (BMI), and arm and calf circumferences (CC). The maximum score is 30 points. A score higher than 23.5 points indicates satisfactory nutritional status, while a lower score indicates malnutrition [22]. Patients were measured and weighed on RADWAG personal weight scales (WPT60 150OW) (RADWAG Balances and Scales, Radom, Poland), waist (WC), hip (HC), arm (AC) and calf (CC) circumferences were measured using SECA measuring tape (SECA Deutschland, Hamburg, Germany). Body mass index (BMI) was calculated by dividing body weight by height in meters squared. Waist to hip ratio (WHR) was calculated as WC divided by hip circumference in centimeters. Waist to height ratio (WHtR) was calculated by the formula: (WC [cm]/height [cm]) × 100. Age, years of education, and chronic diseases of study participants were also recorded.

### 2.3. Statistical Analysis

To detect MNA score difference of 2 points (26 vs. 24) between GDS score groups, with standard deviation of MNA equal to 3 points, 49 subjects are required in each group. To detect an MNA score difference of 2 points (26 vs. 24) between GDS score groups, with standard deviation of MNA equal to 4 points, 86 subjects are required in each group. To detect a correlation coefficient of 0.15 with an alpha of 0.05 and test power of 90.0%, a sample size of 462 individuals is needed. The sample was selected on the base of inclusion and exclusion criteria, and it was a convenience sampling. The normality of data distribution was confirmed by the Shapiro-Wilk test. As the data were non-normally distributed, variables were presented as median values and interquartile ranges, they were compared using the Mann–Whitney U test. The Chi^2^ test was used to compare qualitative values. GDS score and sex were used as grouping variables. Spearman’s and Pearson’s rank correlation coefficients were calculated. For both women and men, independent variables that predicted inclusion to each group with the presence of depression symptoms (GDS > 5) were selected using logistic stepwise regression based on odds ratios and corresponding 95% confidence intervals (95%CI). Variables included in the model were age; years of education; BMI; WC; CC; WHR; WHtR; MNA; presence of hypertension, stroke, cancer, osteoporosis, chronic obstructive pulmonary disease (COPD), congestive heart failure, diabetes, and suffered myocardial infarction. The limit of statistical significance was set at a *p*-value of less than 0.05. Analyses were carried out using Statistica 13.1 software (StatSoft Polska, Cracow, Poland).

### 2.4. Ethical Certification

The study was approved by the Ethics Committee of the Medical University of Lodz (approval number: RNN/73/15/KE), and written informed consent was obtained from all subjects.

## 3. Results

In the study, a total of 2189 patient were examined. Two hundred and fourteen participants were excluded from the study because of missing essential data such as GDS, MNA, education years, or anthropometric variables. Ultimately, 1975 outpatients from the Department of Geriatrics of Medical University in Lodz were enrolled in the study (1457 women and 518 men) (Figure 1).

Table 1 summarizes the general characteristics of the study population according to sex. Statistical analysis revealed no differences in BMI, WHtR, and MNA between women and men. Age, education years, WC, HC, CC, and WHR differed according to sex. The prevalence of depression, osteoporosis, and myocardial infarction differed between women and men (Table 1).

Table 2 shows a comparison of the major anthropometric variables in women and men divided according to the GDS score. In the female group, no differences were found in BMI, WC, HC, and WHR. Women with depression symptoms were older, had received a significantly shorter education, and had a smaller CC and higher WHtR compared to women with a lower GDS score. Significant differences in nutritional status according to the MNA scale were observed. The prevalence of chronic diseases such as depression, hypertension, stroke, COPD, congestive heart failure, diabetes, and myocardial infarction differed according to GDS score. In the male group, there were no differences in age, BMI, WC, HC, WHtR, and WHR between men divided according to GDS score. In the group with depression symptoms, men spent noticeably less time on education, had smaller CC, and scored fewer points on the MNA scale. The prevalence of depression, stroke, and congestive heart failure differed between the GDS groups (Table 2).

Table 3 shows the Spearman’s and Pearson’s rank correlation coefficients of GDS score and major anthropometric variables. Among women education years, CC and MNA negatively correlated with the severity of depression symptoms, and a positive correlation was observed between WHtR and GDS. Among men, GDS negatively correlated with education years, CC, and MNA (Table 3).

Table 4 presents the results of logistic stepwise regression. For both women and men, fewer education years and a lower MNA score were associated with an increased risk of depression symptoms. Additionally, among women, the odds ratio increased with WHtR. Other variables, viz., age, BMI, WC, HC, CC, and WHR were statistically insignificant, as well as chronic diseases. In order to improve visualization of cumulative effects of education years and MNA on GDS score, the results are presented on surface charts (Figure 2 and Figure 3). After excluding 12.2% of people diagnosed with depression, the regression analysis did not change significantly. MNA and level of education remained the only independent significant predictors of depression symptoms. In the women’s group, the relationship with WHtR ceased to be statistically significant, while the relationship with MNA became even stronger.

## 4. Discussion

Obtained results clearly indicate that nutritional status is associated with risk of depression symptoms in the older population. Education level is the second main protective factor, even more important than presence of concomitant diseases. The study included a large representative group of community-dwelling older adults from Central-Eastern Europe. The differences in numbers of women and their age in relation to men reflects demographic situation in Poland and a greater willingness of women to participate in research [17,19]. The difference between the age in women from groups with and without depression symptoms results from the fact that the risk of depression increases with age, which has been noted in many studies [19,23]. Age was included as a disturbing factor in the stepwise regression analysis model without affecting the significance of the analyzed variables.

Previous studies have shown a link between diet and nutritional status assessed with MNA and the risk and severity of depression [24]. However, few of these studies have been conducted among the older population [25,26]. In a recent study conducted in a Greek older population, nutritional status was independently associated with cognitive and psychological status [27]. The combination of an improper diet with low physical activity can lead to sarcopenic obesity through a reduction in muscle mass and an increase in fatty tissue [28,29]. This and other conditions common among older people, such as oedema [30], may interfere with a correct assessment of nutritional status. Diseases that can affect body composition/volume, such as congestive heart failure or immobilization caused by, for example osteoarthritis, merit particular consideration when choosing a suitable nutritional status assessment method in older people groups [31,32]. In such cases, commonly used anthropometric indicators such as BMI, body weight, and circumferences will not be useful [33]. The present study also takes chronic diseases into consideration. Women with depression symptoms are more likely to suffer from hypertension, stroke, congestive heart failure, diabetes, and myocardial infarctions, while men with depression are more susceptible to stroke and congestive heart failure. All statistically significant diseases were included in the regression analysis; however, none of them influenced the significance of educational years and MNA. Only when MNA was excluded from the analysis did the relationship between depression symptoms and congestive heart failure become significant (results not shown in the tables), indicating how strongly nutritional status affects the severity of depression symptoms. The addition of MNA to analysis reduced the significance of other potentially important parameters such as the presence of chronic diseases, which were previously recognized as important risk factors of developing depression symptoms [34]. Another possible interpretation of obtained data may be the fact that depression may reduce appetite and, consequently, increase the risk of malnutrition in older people, which was noted in previous studies [35].

Our findings did not identify any significant correlation between GDS and WHR. The latter should be used to assess the distribution of adipose tissue in obese individuals, not as an indicator of obesity [36]. This finding may suggest that the WHR has minor relevance when assessing older populations. However, in women, a correlation was observed between GDS and WHtR: an anthropometric index for measuring abdominal obesity regardless of sex. WHtR seems to be a better predictor of cardiovascular risk than other indicators [37]. Obtained statistically significant results indicate greater validity of WHtR as an indicator that can be used in older women. However, more research needs to be conducted in order to confirm such a correlation in men. Other commonly used anthropometric variables are circumferences. Previous studies have found larger WC to be associated with increased abdominal adiposity and elevated risk of metabolic syndrome development as well as its consequences such as hypertension or diabetes [38]. WC has been also associated with prevalence of depression symptoms [39]. In the present study, no correlation was found between WC and GDS. As abdominal obesity is associated with a higher prevalence of chronic diseases including depression, there is a need to find the best tool to properly assess fat distribution in older people [40,41]. Smaller CC is associated with frailty syndrome, sarcopenia [42,43], and nutritional status [44]. Our results indicate a significant correlation between CC and GDS in both sexes. Nevertheless, a much stronger correlation was found between MNA and GDS. CC appears to be an important anthropometric variable that may be used to assess nutritional status or decreased muscle mass in the older population, but it is more accurate when combined with other variables and indicators of the MNA scale. It may be especially important given the recently demonstrated high sensitivity of MNA scale for identifying nutritional risk in older adults with COVID−19 [45].

Our results indicate that the severity of depression symptoms in older adults is also associated with lower education level, similarly, as confirmed in previous studies [46]. This highlights the need to focus attention on older persons with lower education level [47]. Likewise, in low-income older adults, increased self-reported depressive symptoms were related to less favorable nutritional status [48]. The stepwise regression analysis confirmed a strong relationship between the severity of depression symptoms, nutritional status assessed with MNA, and number of years of education in the older population. These two predictors are stronger than the accompanying diseases, both in women and men. Worse nutritional status can lead to many disorders, which can promote development of depression symptoms. Mood deterioration can also be associated with improper eating, which in turn, can aggravate nutritional problems and increase the chance of improper dietary choices [49]. This observed relationship between nutritional status and the risk and severity of depression underlines the need for a dietician to participate in the therapeutic process of seniors. Appropriate malnutrition screening can reduce the burden of the health-care system, increase the possibility of faster and more effective treatment introduction, and improve the quality of life of older people. Implementing proper nutrition and compensating for nutritional deficiencies could increase the effectiveness of depression treatment and reduce the severity of its symptoms. Previous studies confirm positive clinical outcomes of dietary supplementation in depressive patients [50]. However, more research is needed to confirm these results worldwide.

The main advantages of our study are careful recruitment of patients and large group size. So far, no study has been conducted to bind nutritional status assessment tools and depression symptoms in such a big population of older adults. It also uses internationally recognized tests for assessing nutritional status (MNA) and severity of depression symptoms (GDS), both of which have been properly validated in Poland. Furthermore, anthropometric measurements were performed with great care with the use of validated professional medical equipment. However, there are some limitations. Firstly, the study group was restricted to Central-European community-dwelling older individuals therefore results may be different in other cultures. Correlations of nutritional status and depression may also be different in the institutionalized environments [51,52]. Secondly, no tests were conducted among people with severe dementia or those who were not able to present at the clinic due to mobility problems. Thirdly, seasonal variations of mood could also be important. Finally, the most important aim of this study was to analyze the most common anthropometric variables, which are easy to use and widely available in healthcare premises. Nevertheless, future research should corroborate present findings with more exact body composition analyses such as bioimpedance or DEXA.

Further research should be considered to evaluate the impact of diet and nutritional status on depression among older patients and to confirm the need for comprehensive nutritional assessments performed by medical professionals in routine treatment and follow up. Intervention and prospective cohort studies should also be considered to verify whether appropriate nutritional intervention and improvement in nutritional status influence the severity of depression symptoms in the older populations.

## 5. Conclusions

Our findings indicate a strong relationship between nutritional status assessed with the MNA scale, education years, and the severity of depression symptoms in both women and men. WHtR was also associated with the presence of depression symptoms in women.

## Figures and Tables

**Figure 1 nutrients-13-00515-f001:**
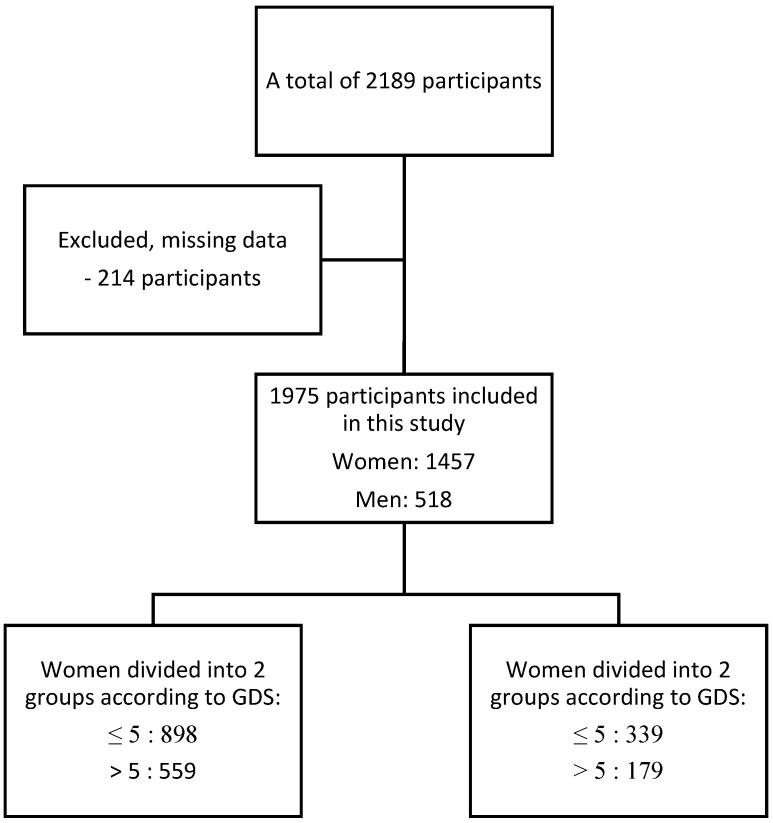
Flowchart of the study.

**Figure 2 nutrients-13-00515-f002:**
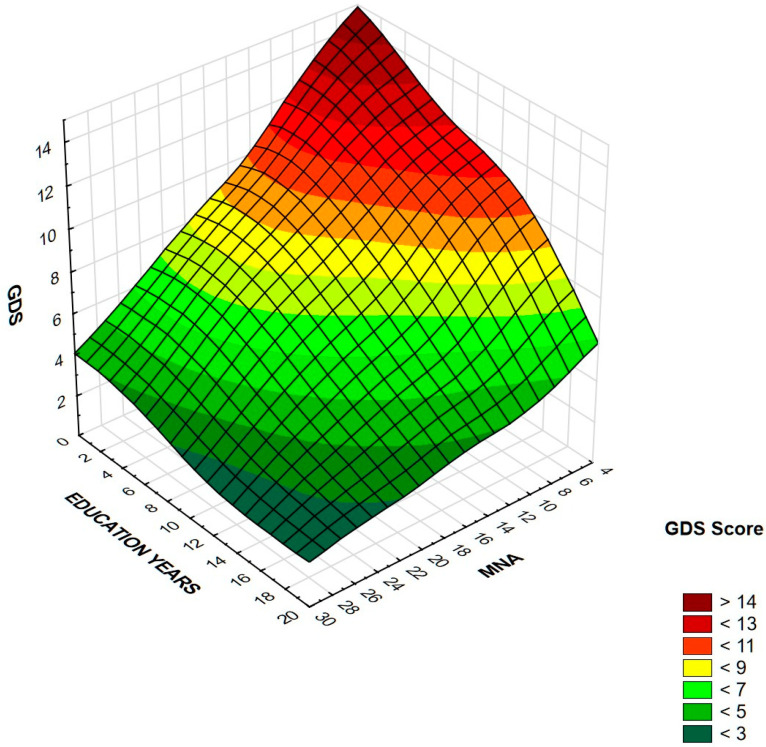
Association between GDS score, education years, and MNA in women. Higher GDS is associated with lower education level and lower MNA score in this group.

**Figure 3 nutrients-13-00515-f003:**
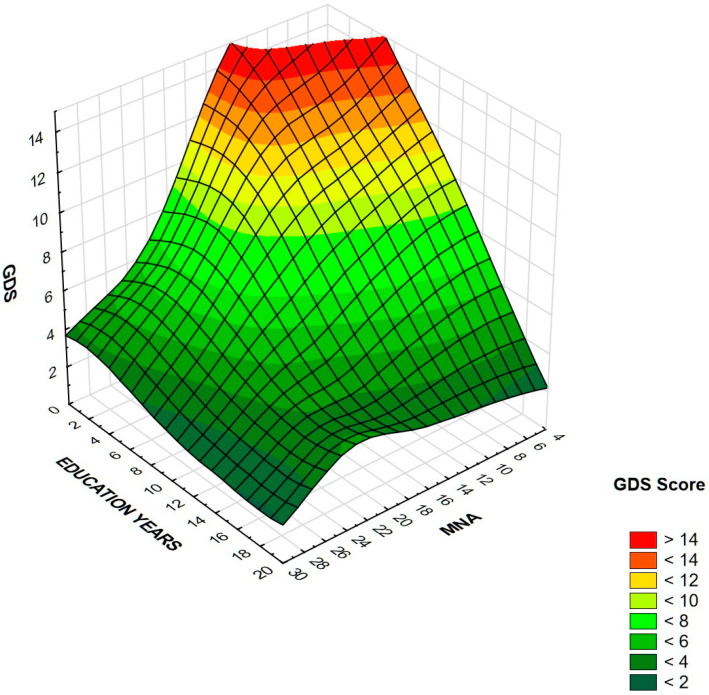
Association between GDS score, education years, and MNA in men. Higher GDS is associated with lower education level and lower MNA score in this group.

**Table 1 nutrients-13-00515-t001:** General characteristic of the study population (*n* = 1975) according to sex.

Variable	All (*n* = 1975)	Women (*n* = 1457)	Men (*n* = 518)	*p*-Value
Age [years]	75 (67–80)	75 (67–81)	73 (66–78)	<0.001 ^a^
Education [years]	11 (7–14)	11 (7–13)	11 (7–15)	0.023 ^a^
BMI [kg/m^2^]	26.9 (24.1–30.1)	26.9 (23.9–30.5)	26.7 (24.5–29.5)	ns ^a^
Waist circumference [cm]	93 (84–102)	91 (82–100)	99 (92–107)	<0.001 ^a^
Hips circumference [cm]	104 (98–111)	105 (98–112)	102 (98–107)	<0.001 ^a^
Calf circumference [cm]	36 (33–38)	35 (33–38)	36 (34–39)	<0.001 ^a^
WHtR	57.9 (52.6–63.6)	58.1 (52.2–63.7)	57.7 (53.9–62.9)	ns ^a^
WHR	0.89 (0.83–0.94)	0.86 (0.81–0.91)	0.97 (0.93–1.01)	<0.001 ^a^
MNA	25.5 (22.5–27.5)	25.5 (22.5–27.5)	25.7 (23.5–27.5)	ns ^a^
Depression [*n* (%)]	241 (12.2)	196 (13.4)	45 (8.7)	0.004 ^b^
Hypertension [*n* (%)]	1330 (67.3)	995 (68.2)	335 (64.7)	ns ^b^
Stroke [*n* (%)]	233 (11.8)	167 (11.5)	66 (12.7)	ns ^b^
Cancer [*n* (%)]	165 (8.4)	118 (8.1)	47 (9.1)	ns ^b^
Osteoporosis [*n* (%)]	485 (24.6)	403 (27.7)	82 (15.8)	<0.001 ^b^
COPD [*n* (%)]	114 (5.8)	86 (5.9)	28 (5.4)	ns ^b^
Congestive heart failure [*n* (%)]	745 (37.7)	533 (36.6)	212 (40.9)	ns ^b^
Diabetes [*n* (%)]	376 (19.1)	266 (18.3)	110 (21.3)	ns ^b^
Myocardial infarction [*n* (%)]	214 (10.8)	134 (9.2)	80 (15.4)	<0.001 ^b^

The quantitative values are presented as median and interquartile difference, qualitative values as number and percentage. ^a^ Mann–Whitney U-test; ^b^ Chi^2^-test. Abbreviations: BMI—body mass index, WHtR—waist to height ratio, WHR—waist to hip ratio, MNA—Mini Nutritional Assessment, GDS—Geriatric Depression Scale, COPD—chronic obstructive pulmonary disease.

**Table 2 nutrients-13-00515-t002:** Comparison of anthropometric variables, MNA, and prevalence of chronic diseases in female and male groups divided according to GDS score (GDS ≤ 5 vs. GDS > 5 separately in women and men).

Variable	Women GDS ≤ 5 (*n* = 898) Median (Quartiles)	Women GDS > 5 (*n* = 559) Median (Quartiles)	Men GDS ≤ 5 (*n* = 339) Median (Quartiles)	Men GDS > 5 (*n* = 179) Median (Quartiles)
Age [years] ^a^	74 (66–79)	77 (70–83) ***	74 (66–79)	72 (68–77)
Education [years] ^a^	12 (8–16)	7 (7–12) ***	12 (9.5–16)	10 (7–12) ***
BMI [kg/m2] ^a^	27 (24.2–30.1)	26.7 (23.4–30.1)	27 (24.6–29.7)	26.5 (24.1–29.4)
Waist circumference [cm] ^a^	90 (82–99)	91 (82–102)	99 (93–106)	100 (90–108)
Hips circumference [cm] ^a^	105 (99–112)	105 (98–114)	102 (98–107)	102 (96–107)
Calf circumference [cm] ^a^	36 (33–38)	34 (32–37) ***	57.2 (54–62)	36 (33–38) *
WHtR ^a^	57.4 (52.3–62.9)	58.8 (52.1–65.6) *	57.2 (54–62)	58.7 (53–63.8)
WHR ^a^	0.86 (0.81–0.91)	0.87 (0.82–0.91)	0.97 (0.93–1)	0.96 (0.93–1)
MNA ^a^	26.5 (24–28)	23 (20.5–26) ***	26.5 (24.5–28)	24 (20.5–26.5) ***
Depression [*n* (%)] ^b^	107 (11.9)	89 (15.9) *	17 (5)	28 (15.6) ***
Hypertension [*n* (%)] ^b^	583 (64.9)	412 (73.7) ***	218 (64.3)	117 (65.4)
Stroke [*n* (%)] ^b^	70 (7.8)	97 (17.3) ***	31 (9.1)	35 (19.5) ***
Cancer [*n* (%)] ^b^	76 (8.5)	42 (7.5)	29 (8.5)	18 (10.1)
Osteoporosis [*n* (%)] ^b^	243 (27.1)	160 (28.6)	57 (16.8)	25 (14)
COPD [*n* (%)] ^b^	62 (6.9)	24 (4.3) *	17 (5)	11 (6.2)
Congestive heart failure [*n* (%)] ^b^	291 (32.4)	242 (43.3) ***	122 (36)	90 (50.3) **
Diabetes [*n* (%)] ^b^	141 (15.7)	125 (22.4) ***	72 (21.2)	38 (21.3)
Myocardial infarction [*n* (%)] ^b^	71 (7.9)	63 (11.3) *	50 (14.7)	30 (16.8)

The quantitative values are presented as median and interquartile difference, qualitative values as number and percentage. ^a^ Mann-Whitney U-test; ^b^ Chi^2^-test; * *p* ≤ 0.05; ** *p* ≤ 0.01; *** *p* ≤ 0.001; Abbreviations: GDS—Geriatric Depression Scale, BMI—body mass index, WHtR—waist to height ratio, WHR—waist to hip ratio, MNA—Mini Nutritional Assessment, COPD—chronic obstructive pulmonary disease.

**Table 3 nutrients-13-00515-t003:** Spearman’s and Pearson’s rank correlation coefficients between GDS score and education years, high, weight, BMI, waist, hip and calf circumferences, WHR, WHtR, and MNA in women and men.

Parameters	Women	Men
	rS (rP)	rS (rP)
Age [years]	0.25 *** (0.22 ***)	−0.02 (−0.04)
Education [years]	−0.37 *** (−0.36 ***)	−0.31 *** (−0.30 ***)
BMI [kg/m^2^]	−0.03 (0.00)	−0.06 (−0.06)
Waist circumference [cm]	0.04 (0.04)	−0.001 (−0.04)
Hips circumference [cm]	0.04 (0.05)	−0.004 (0.003)
Calf circumference [cm]	−0.17 *** (−0.13 ***)	−0.13 ** (−0.11 **)
WHR	0.03 (−0.01)	−0.03 (−0.07)
WHtR	0.06 * (0.05 *)	−0.02 (0.002)
MNA	−0.42 *** (−0.38 ***)	−0.40 *** (−0.45 ***)

rS—Spearman’s rank correlation, rP—Pearson’s rank correlation, WHtR—waist to height ratio, MNA—Mini Nutritional Assessment * *p* ≤ 0.05; ** *p* ≤ 0.01; *** *p* ≤ 0.001.

**Table 4 nutrients-13-00515-t004:** Odds ratios (95% confidence interval) for the risk of belonging to the group with higher score of depression symptoms (GDS > 5) for education years, MNA score, and WHtR.

Variable	Women			Men		
	OR	95%CI	*p*	OR	95%CI	*p*
Education [years]	0.87	0.84–0.90	<0.001	0.9	0.85–0.95	<0.001
MNA	0.83	0.80–0.86	<0.001	0.81	0.75–0.87	<0.001
WHtR	1.02	1.002–1.03	0.03	-	-	-

OR—odds ratio, 95%CI—95% confidence interval.

## Data Availability

Not applicable.

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
