# Peer review of "Association of Lower Nutritional Status and Education Level with the Severity of Depression Symptoms in Older Adults—A Cross Sectional Survey"

_nutrients, 2021, doi:10.3390/nu13020515_

Round 1
Reviewer 1 Report
This is a timely study and very appropriate. It has a good scientific validity. It was well written and the analysis was done to explore important factors. In fact the two figures are very interesting and provide new concept of visualising important factors.
Following factors need to be clarified.
Abstract Line 34- MNA repeating
Method - Need to add sample size calculation, sampling
Discussion -
Line 239-240 - It is not sure how representative this sample. Details are not provided in the method section. This sample is not community dwelling. This is a clinic based sample.
Line 241-242 - According to the demographic profile of Poland male: female ratio between 55-64 years and >=65 years are 0.9 and 0.66 respectively. In the current study male:female ratio is 0.36. It is better to explain it in a valid way. Reference : https://www.indexmundi.com/poland/demographics_profile.html
Line 333-334 - Recruitment process is not given in the method to verify this statement.
Author Response
Reviewer 1
This is a timely study and very appropriate. It has a good scientific validity. It was well written and the analysis was done to explore important factors. In fact the two figures are very interesting and provide new concept of visualising important factors.
Thank you very much for the very valuable review.
Following factors need to be clarified.
Abstract Line 34- MNA repeating
Has been corrected
Method - Need to add sample size calculation, sampling
Has been added
Discussion -
Line 239-240 - It is not sure how representative this sample. Details are not provided in the method section. This sample is not community dwelling. This is a clinic based sample.
The patients participating in the study are community dwelling older adults. Their visits to the Geriatrics clinic are conditioned not only by health control visit but also participation in the research projects conducted in our clinic. This information has been added to the materials and methods section.
Line 241-242 - According to the demographic profile of Poland male: female ratio between 55-64 years and >=65 years are 0.9 and 0.66 respectively. In the current study male:female ratio is 0.36. It is better to explain it in a valid way. Reference : https://www.indexmundi.com/poland/demographics_profile.html
The information about demographic profile of Poland, where the ratio of male to female is 0.66 ≥65 years of age have been added. We choose to add only this information because median age of our patients is 75. We also add information about possible factor influencing the numerical superiority of women over men on the base of the POLSENIOR study.
Reference: Bledowski, P.; Mossakowska, M.; Chudek, J.; Grodzicki, T.; Milewicz, A.; Szybalska, A.; Wieczorowska-Tobis, K.; Wiecek, A.; Bartoszek, A.; Dabrowski, A., et al. Medical, psychological and socioeconomic aspects of aging in Poland: assumptions and objectives of the PolSenior project. Exp Gerontol 2011, 46, 1003-1009, doi:10.1016/j.exger.2011.09.006.
Line 333-334 - Recruitment process is not given in the method to verify this statement.
Information about the recruitment process, number of excluded participants have been added into the Materials and Methods section.
Reviewer 2 Report
Summary: This study analyzes the relationship between nutritional status and depression symptom severity among older adults in Poland. Depression symptom severity was assessed using GDS. The nutritional status of patients was evaluated using MNA and anthropometry. Demographic and health variables were also measured. The results indicate strong protective effects of proper nutritional status and education level against developing depression among adults over 60yrs.
Comments
1. The authors need a native English speaker to edit the paper. Grammar is incorrect or awkward in a few places - mostly in the introduction. It could be improved. For example: “Recent decades have seen a growth in the ageing of societies”. This would make more sense as: “Recent decades have seen an increase in the proportion of older adults due to extended lifespans.” Please have a native English speaker edit.
2. Methods: Please state/explain how the sample was selected (e.g. purposefully selected, convenience sample, random sample). If the sample was randomly selected that will strengthen the findings. Sorry if I missed where it is stated.
3. Statistical analysis: On what assumptions was the sample size calculated? How did you arrive at this large sample size? Was it based on finding a certain sizeable difference in one specific outcome variable? We assume the primary purpose was to investigate the size of difference between MNA by GDS score but what sizeable difference was expected? Please state the power calculations. The primary outcome was to examine the relationship between nutritional status and the severity of depression symptoms.
4. Table 1: I assume there are no statistical differences. Please state this in the paragraph describing these results. Also, within the table, please correct the (n, %) to reflect what is actually written: n (%)
5. Table 2: Please think about what are the meaningful differences here for the reader to focus on (if you can) and don’t focus on the differences that appear significant only because you have such a large sample size. Please correct the n (%) within the table. It might be more accurately represented by using [n (%)].
6. Please give a description of the trends found in figs 1 & 2. Higher GDS is associated with lower education and higher MNA...
Good luck.
Author Response
Reviewer 2
Summary: This study analyzes the relationship between nutritional status and depression symptom severity among older adults in Poland. Depression symptom severity was assessed using GDS. The nutritional status of patients was evaluated using MNA and anthropometry. Demographic and health variables were also measured. The results indicate strong protective effects of proper nutritional status and education level against developing depression among adults over 60yrs.
Thank you very much for the very valuable review.
Comments
1. The authors need a native English speaker to edit the paper. Grammar is incorrect or awkward in a few places - mostly in the introduction. It could be improved. For example: “Recent decades have seen a growth in the ageing of societies”. This would make more sense as: “Recent decades have seen an increase in the proportion of older adults due to extended lifespans.” Please have a native English speaker edit.
The first version of the manuscript was checked by a native speaker, however, as requested by the reviewer, it was re-sent for proofreading,
- Methods: Please state/explain how the sample was selected (e.g. purposefully selected, convenience sample, random sample). If the sample was randomly selected that will strengthen the findings. Sorry if I missed where it is stated.
The sample was selected on the base of the inclusion and exclusion criteria and it was a convenience sampling. This information has been added into the material and methods section.
- Statistical analysis: On what assumptions was the sample size calculated? How did you arrive at this large sample size? Was it based on finding a certain sizeable difference in one specific outcome variable? We assume the primary purpose was to investigate the size of difference between MNA by GDS score but what sizeable difference was expected? Please state the power calculations. The primary outcome was to examine the relationship between nutritional status and the severity of depression symptoms.
To detect the MNA difference of 2 points (26 vs 24) between GDS score groups, with standard deviation of MNA equal to 3 points, 49 subjects are required in each group. To detect the MNA difference of 2 points (26 vs 24) between GDS score groups, with standard deviation of MNA equal to 4 points, 86 subjects are required in each group. To detect correlation coefficient of 0.15 with alpha of 0.05 and power of 90.0%, sample size of 462 for each group is needed (924 in total). The sample was selected on the base of inclusion and exclusion criteria and it was a convenience sampling. This information has been added into the material and method section.
- Table 1: I assume there are no statistical differences. Please state this in the paragraph describing these results. Also, within the table, please correct the (n, %) to reflect what is actually written: n (%)
There are statistical differences, unfortunately the reviewers received a distorted table due to the formatting error in the journal. The text has been reformatted and the tables have been corrected so that all data is properly visible. We changed ‘(n,%)’ into the [n (%)] in every table as requested.
- Table 2: Please think about what are the meaningful differences here for the reader to focus on (if you can) and don’t focus on the differences that appear significant only because you have such a large sample size. Please correct the n (%) within the table. It might be more accurately represented by using [n (%)].
All the data shown in the tables are meaningful because they enable us to characterize the group. Differences obtained in the analyzes determine which data will be used for subsequent analyzes (in this case – correlations and stepwise regression).
- Please give a description of the trends found in figs 1 & 2. Higher GDS is associated with lower education and higher MNA...
Description of the trends found in figures 1and 2 has been added.
Good luck.
Reviewer 3 Report
Overall
This is a good paper on an important topic. The nutritional needs of adults are too easily ignored by policy-makers and practitioners in HIC despite this having clear potential to adversely affect health, wellbeing and mental health (as is the focus of this paper).
Novelty and direct added value to the literature in this area should be commented on by a topic-specific expert but seems to me that the paper makes a useful contribution to the literature and should be published. It would benefit from some changes prior to publication:
Title:
- This should include the study type : a cross-sectional survey
- “cumulative” should not be used in title as it can be misread to imply a (stronger) longitudinal design. Better to include the phrase “associations between…” – since this is what the project directly examined
Abstract
- Final sentence should be edited to reflect the data: all that’s been shown is an association, causation has not been shown. Direction of association also could be the opposite: depression leading to poor nutrition. Whilst I share the authors’ belief in the likely benefits of improving nutrition, this needs to be demonstrated in a future study before this claim can be made so directly (the need for this future work could be highlighted)
Background
- Would be clearer to differentiate:
- ONE overall study aim (ie a higher level-ambition, such as to inform policy/practice in this topic)
- Specific objectives - as described
Methods
- Please say more about the study participants. Outpatients for what problems/diagnoses? How were they identified/selected/recruited?
- Please edit and expand on lines 93-95.
- From other sources, though older females do outnumber older males in Poland, this almost 3:1 female skew in numbers does not seem to be at all representative of the wider Polish population https://www.populationpyramid.net/poland/2019/
- It is also very unlikely that a population of hospital outpatients are representative of the wider population as is implied by sentence as currently phrased.
Improving nutrition/mental health of elderly outpatients is of course itself an important aim, but this should be acknowledged in the discussion/limitations. One cannot extrapolate to the general population as authors seem to be doing in this paper.
- Line 96 – the exclusions are sensible – but there needs to be a flow chart in the results to show how many were excluded and for which reasons. (related to this, editors should check STROBE checklist to ensure all items have been reported)
- Has GDS been validated in Poland? Are the cutoffs of <5 as valid there as in other settings?
- Line 154 – please give ethics reference number
Results
- As noted previously, a study flow chart should be included in the first section of results
- Over what period/dates were the participants enrolled/assessed? (if a long period might there be some seasonal effects…e.g. higher rates of depression in winter months?)
- Tables should be better formatted so that each variable is on one line and that the columns are narrower to allow easier reading and comparison of male/female differences
- Tables 2,3 are presumably unadjusted comparisons?
- Since it’s important to recognise male/female differences, could authors combine tables 2 and 3 by having ONE table with more columns – this could make it easier to compare across the two.
- In tables 2,3 there might also usefully be a column showing what the differences are (e.g. mean difference in age/education) – this of course can be derived, but a value with an associated confidence interval would help readers understand key data more quickly and easily and would help then consider to what extent a statistically significant difference is also clinically significant.
- It’s interesting to note higher prevalence of some disease conditions such as hypertension, stroke. Is any other data available on underlying medical history? Seems like there might be other (unmeasured or unreported) morbidity differences which could confound the association between GDS and nutrition – this should be discussed in limitations section of discussion.
- Figures 1 and 2 work well and are a helpful way of visualizing the data.
Discussion
Overall well written with correct and appropriate focus on “associations” and not over-reaching to infer causation. A few things might help strengthen the discussion:
- Noting any evidence on whether depression in elderly populations leads to decreased appetite/intake and hence malnutrition (which of course is the other possible interpretation of this date)
- Using Bradford-Hill criteria to frame the discussion and strengthen the arguments that lower nutritional status can predispose to depression symptons
- Citing any intervention studies that involve nutrition support/supplementation and examining impact on depression (to complement/precede the discussion from line 326-332) I’m not familiar with the literature in this area but would be surprised if at least some such studies hadn’t been done.
- Limitations section should be expanded to better highlight that this was a population of hospital outpatients. May not even be applicable to the general older population in the same setting let alone in other cultures (Though equally the association could be even stronger, especially in resource-poor settings where background nutritional status is much worse)
- Conclusion is OK, but might be better to be slightly shorter and have recommendations for future research as a slightly expanded stand-alone paragraph just before the conclusion (recommendations should also include an intervention study as well as prospective cohort studies)
Author Response
Reviewer 3
This is a good paper on an important topic. The nutritional needs of adults are too easily ignored by policy-makers and practitioners in HIC despite this having clear potential to adversely affect health, wellbeing and mental health (as is the focus of this paper).
Novelty and direct added value to the literature in this area should be commented on by a topic-specific expert but seems to me that the paper makes a useful contribution to the literature and should be published. It would benefit from some changes prior to publication:
Thank you very much for the very valuable review.
Title:
- This should include the study type : a cross-sectional survey
Has been added to the title.
- “cumulative” should not be used in title as it can be misread to imply a (stronger) longitudinal design. Better to include the phrase “associations between…” – since this is what the project directly examined
“Association of lower nutritional status and education level with the severity of depression symptoms in older adults – a cross-sectional survey.” has been corrected.
Abstract
- Final sentence should be edited to reflect the data: all that’s been shown is an association, causation has not been shown. Direction of association also could be the opposite: depression leading to poor nutrition. Whilst I share the authors’ belief in the likely benefits of improving nutrition, this needs to be demonstrated in a future study before this claim can be made so directly (the need for this future work could be highlighted)
Has been changed due to the reviewer’s suggestion, both in the abstract and conclusions.
Background
- Would be clearer to differentiate:
- ONE overall study aim (ie a higher level-ambition, such as to inform policy/practice in this topic)
- Specific objectives - as described
Has been changed according to the reviewer’s suggestion.
Methods
- Please say more about the study participants. Outpatients for what problems/diagnoses? How were they identified/selected/recruited?
Study participants were community-dwelling older adults. Their visits in the Geriatric Clinic were conditioned not only by health control visits but also participation in the research projects conducted in the clinic. The sample was selected on the basis of inclusion and exclusion criteria and it was a convenience sampling. This information has been added in the Material and Method section.
- Please edit and expand on lines 93-95.
- From other sources, though older females do outnumber older males in Poland, this almost 3:1 female skew in numbers does not seem to be at all representative of the wider Polish population https://www.populationpyramid.net/poland/2019/
- It is also very unlikely that a population of hospital outpatients are representative of the wider population as is implied by sentence as currently phrased.
The information about demographic profile of Poland, where the ratio of male to female is 0.66 ≥65 years of age have been added. We choose to add only this information because median age of our patients is 75. We also add information about possible factor influencing the numerical superiority of women over men on the base of the POLSENIOR study.
Reference: Bledowski, P.; Mossakowska, M.; Chudek, J.; Grodzicki, T.; Milewicz, A.; Szybalska, A.; Wieczorowska-Tobis, K.; Wiecek, A.; Bartoszek, A.; Dabrowski, A., et al. Medical, psychological and socioeconomic aspects of aging in Poland: assumptions and objectives of the PolSenior project. Exp Gerontol 2011, 46, 1003-1009, doi:10.1016/j.exger.2011.09.006.
Our patients come to the clinic to undergo a standard health examination / medical consultation or to take part in the project conducted at our University. They are a community-dwelling older adults.
Improving nutrition/mental health of elderly outpatients is of course itself an important aim, but this should be acknowledged in the discussion/limitations. One cannot extrapolate to the general population as authors seem to be doing in this paper.
- Line 96 – the exclusions are sensible – but there needs to be a flow chart in the results to show how many were excluded and for which reasons. (related to this, editors should check STROBE checklist to ensure all items have been reported)
In the study total of 2189 patients were examined. Two hundred and fourteen participants were excluded from the study because of missing essential data such as GDS, MNA, education years, or anthropometric variables. This information has been added and all of this information was moved to the results section with the flow chart required by the reviewer.
- Has GDS been validated in Poland? Are the cutoffs of <5 as valid there as in other settings?
The Geriatric Depression Scale is a standard questionnaire used to assess the severity of depression symptoms in Poland. It was used, among others, in the PolSenior project, carried out in 2007-2012, the largest gerontology research project in Poland and one of the largest in Europe. In the PolSenior study clinically significant depression symptoms were diagnosed when the score of GDS was >5.
Reference: Bledowski, P.; Mossakowska, M.; Chudek, J.; Grodzicki, T.; Milewicz, A.; Szybalska, A.; Wieczorowska-Tobis, K.; Wiecek, A.; Bartoszek, A.; Dabrowski, A., et al. Medical, psychological and socioeconomic aspects of aging in Poland: assumptions and objectives of the PolSenior project. Exp Gerontol 2011, 46, 1003-1009, doi:10.1016/j.exger.2011.09.006.
- Line 154 – please give ethics reference number
Has been added
Results
- As noted previously, a study flow chart should be included in the first section of results
Has been added.
- Over what period/dates were the participants enrolled/assessed? (if a long period might there be some seasonal effects…e.g. higher rates of depression in winter months?)
As a matter of fact, patients were enrolled during several years on different seasons. Unfortunately, data on eventual seasonal variations are not available in the present study. This possibility has been mentioned in the discussion/limitations.
- Tables should be better formatted so that each variable is on one line and that the columns are narrower to allow easier reading and comparison of male/female differences
Unfortunately, the reviewers received a distorted table due to the formatting error in the journal. The text has been reformatted and the tables have been corrected so that all data is properly visible.
- Tables 2,3 are presumably unadjusted comparisons?
Yes, the comparisons are unadjusted.
- Since it’s important to recognise male/female differences, could authors combine tables 2 and 3 by having ONE table with more columns – this could make it easier to compare across the two.
- In tables 2,3 there might also usefully be a column showing what the differences are (e.g. mean difference in age/education) – this of course can be derived, but a value with an associated confidence interval would help readers understand key data more quickly and easily and would help then consider to what extent a statistically significant difference is also clinically significant.
Thank you for this valuable suggestion. Tables 2 and 3 were combined as suggested by the reviewer. It is difficult to add reference values to such a table because they are different for men and women and this could make the table unreadable. In addition, only 5 out of 18 variables presented in the table have reference values presented by WHO. Therefore, the comparison between women and men was presented in Table 1 and that between GDS groups separately in women and men in Table 2.
- It’s interesting to note higher prevalence of some disease conditions such as hypertension, stroke. Is any other data available on underlying medical history? Seems like there might be other (unmeasured or unreported) morbidity differences which could confound the association between GDS and nutrition – this should be discussed in limitations section of discussion.
Yes, there may be some other unmeasured or unreported morbidity that may affect nutritional status or GDS, but the relationship between GDS and MNA shown in the study is so strong that such possibility is rather low. The most common chronic diseases were included in the study. Less common diseases that may occur in the population would be rather individual and would not affect the statistical analysis in such a large group. The prevalence of diseases is comparable with other large studies from Poland, e.g. PolSenior study.
- Figures 1 and 2 work well and are a helpful way of visualizing the data.
We really appreciate this valuable opinion.
Thank you very
Discussion
Overall well written with correct and appropriate focus on “associations” and not over-reaching to infer causation. A few things might help strengthen the discussion:
- Noting any evidence on whether depression in elderly populations leads to decreased appetite/intake and hence malnutrition (which of course is the other possible interpretation of this date)
Thank you for this suggestion it has been added in the discussion as other possible interpretation of obtained data.
- Using Bradford-Hill criteria to frame the discussion and strengthen the arguments that lower nutritional status can predispose to depression symptoms
Thank you so much for recalling these criteria, we have made every effort to ensure that the discussion complies with the Sir Bradford-Hill criteria wherever possible.
- Citing any intervention studies that involve nutrition support/supplementation and examining impact on depression (to complement/precede the discussion from line 326-332) I’m not familiar with the literature in this area but would be surprised if at least some such studies hadn’t been done.
It is also very valuable suggestion according to which we have added the appropriate citation in the place indicated by the reviewer.
- Limitations section should be expanded to better highlight that this was a population of hospital outpatients. May not even be applicable to the general older population in the same setting let alone in other cultures (Though equally the association could be even stronger, especially in resource-poor settings where background nutritional status is much worse).
The patients participating in the study were community-dwelling older adults as explained in the previous answers to the reviewer. Limitations section has also been expanded to better present this issue.
- Conclusion is OK, but might be better to be slightly shorter and have recommendations for future research as a slightly expanded stand-alone paragraph just before the conclusion (recommendations should also include an intervention study as well as prospective cohort studies)
As suggested, we shortened the conclusion and moved future research recommendations as a separate paragraph before the conclusions.
Reviewer 4 Report
I don't think this manuscript is worth being published.
Major methodological flaws, lack of novelty
Author Response
Reviewer 4
I don't think this manuscript is worth being published.
Major methodological flaws, lack of novelty
Thank you for your review, we are sorry that you could not justify your opinion in more detail.
Round 2
Reviewer 4 Report
I think manuscript really improved now, so after this thorough revision, it is worth being published.